# The Profile of Bullying Perpetrators and Victims and Associated Factors among High School Learners in Tshwane District, South Africa

**DOI:** 10.3390/ijerph20064916

**Published:** 2023-03-10

**Authors:** Dudu Shiba, Kebogile Elizabeth Mokwena

**Affiliations:** 1Department of Public Health, School of Health Care Sciences, Sefako Makgatho Health Sciences University, Medunsa 0204, South Africa; 2Substance Abuse and Population Mental Health, Sefako Makgatho Health Sciences University, Medunsa 0204, South Africa

**Keywords:** peer violence, perpetrators, bully victims, high school learners, South Africa

## Abstract

Although bullying in South African schools remains a current public health and education discussion, the view has been limited to acts of criminality, and not much has been done to identify risk factors for being bullying perpetrators and victims in a school environment. This study used a cross sectional quantitative survey to determine the profile of bullying perpetrators and victims among high school learners in a township in Pretoria. The Illinois Bully Scale was used to screen for bullying perpetration and victimization, whilst the Patient Health Questionnaire-9 and the Beck Anxiety Inventory were used to screen for depression and anxiety symptoms, respectively, among the sample of learners. STATA version 14 was used for data analysis. The sample of 460 consisted of 69% females with a mean age of 15 years. The 73.91% of learners who fitted the categories of bullying consisted of 21.96% victims, 9.57% perpetrators, and 42.39% perpetrator–victims. The Pearson Chi^2^ test of association found a significant association between being a bullying victim and reported lack of people who loved and cared for the learner. Being a bullying perpetrator was associated with anxiety symptoms of the learner and home alcohol use, while being a perpetrator –victim was associated with lack of family love and care, the school attended, as well as depression and anxiety symptoms. Using multivariate logistic regression, being a perpetrator–victim was associated with depression symptoms, anxiety symptoms, and home use of alcohol whilst being a perpetrator was associated with lack of anxiety symptoms. The study concluded that anxiety, depression, and the home environment are strongly associated with bullying, and most learners fitted the category of being both perpetrators and victims.

## 1. Introduction

Bullying, which can occur verbally, physically, and/or emotionally is abusive repetitive or likely to be repeated behavior, perpetrated on a person or group of people where there is real or perceived power imbalance in favor of the perpetrator [1]. Bullying takes many forms, from direct physical harm, to verbal teasing and threats, to exclusion, humiliation, and spreading malicious rumors about another person [2]. In recent years, a new form of bullying called cyberbullying has emerged, which occurs when technology is deliberately and repeatedly used to bully, harass, hassle, and threaten others, and often the victim is left without any escape [3]. On the other hand, the emergence of cyberbullying has challenged the earlier definition of bullying as electronic messages would not require a bully to repeat the messages but these can spread widely as other people share and forward it. However, there is international consensus on the inclusion of cyberbullying in the definition of bullying, which was adopted by the Center for Disease Control and Prevention [1].

Global literature reports high levels of bullying victimization among high school learners [4,5,6]. The prevalence varies across regions and countries and communities, and while a high prevalence of 74% was reported in Samoa [4], the average prevalence for the Sub Saharan Africa region countries was 38.8% [7]. International and regional studies have found the prevalence of bullying perpetration to be lower than that of victimization, with the prevalence ranging between 1% and 36% [5,8]. In South Africa, reported prevalence of bullying victimization varied from 16.5% [9] to 75% [10], and that of perpetration varied from 3.9% [9] to 8.2% [11]. A previous study that examined bullying victimization in Tshwane District found a prevalence rate of 53.1% [12]. However, this study was conducted more than two decades ago.

Mental disorders are among the key global drivers of mortality and morbidity [13,14]. Literature reports mental health challenges, such as anxiety, depression, and posttraumatic stress, among victims and perpetrators of bullying [4,15,16,17,18,19,20]. The three mental health problems have also been found to be the most common among children and adolescents in South Africa [21]. Because bullying is among the most common forms of mental and physical assault among children, it needs to be addressed at the legal and policy level [22].

The association between bullying and a number of negative health and developmental outcomes that transcend over future generations makes bullying one of the key determinants of health [23,24] and therefore a significant public health problem. Given the widespread prevalence of bullying behavior in schools, its many harmful outcomes on the victim, the bully, and society, its inter-generational impact, its association with other problematic behaviors, and the growing evidence of interventions to address it, bullying requires the attention at policy levels [22] as it frustrates the promotion of child health and development.

Although being a bullying perpetrator and being a victim were traditionally considered to be mutually exclusive, emerging literature has identified a category of individuals fitting as both perpetrators and victims, which is determined by the power relations in a given situation. This category of bullying behavior has also caught the attention of social behavior scientists and is integral to understanding bullying in a comprehensive way.

Being a school bullying perpetrator has been associated with a range of anti-social behaviors, which include violent behavior, delinquency, theft, risk taking behavior [11,25,26]. Bullying perpetrators have also been associated with alcohol and drug use and anxiety [15,27], carrying a weapon to school [28], dropping out of school [29,30], increased odds of having multiple sex partners [15], and criminality [11]. On the other hand, being a victim of bullying has been associated with mental health problems, especially depression and anxiety, having fewer friends, and poor academic achievement [31,32], as well as compromised social skills, which often result in reduced adaptation to adult roles, reduced ability to form lasting relationships, integrating into work, and being economically independent [33]. Parental alcohol symptoms have been associated with bullying perpetration among children [34]. Perceptions of being loved and supported by family, friends, and significant others, including teachers, have been found to be protective factors for being bullied among children [16,35], whilst lower levels of parental support have been associated with higher rates of bullying victimization [36]. Gender has also been found to play a role in bullying among high school learners where being a male learner has been associated with both bullying victimization and bullying perpetration [11,35,37,38,39].

Despite the huge burden of mental distress and anti-social behaviors associated with bullying, as well as the risks that bullying poses on the current and future health and social well-being of victims and perpetrators, bullying behavior is often viewed erroneously as only a matter or ill-discipline of the perpetrators and as a normal rite of passage [33], and does not receive the attention it deserves. In South Africa, policy measures to address bullying have mostly adopted a criminalizing and punitive approach [40], with little focus on addressing the intrinsic drivers of bullying at the individual level as well as the contextual drivers that underlie the problem of bullying. Moreover, there is limited literature on the contributory factors to bullying, as well as its impact on mental and social health in low and middle income countries such as South Africa.

Addressing bullying among young people is one of the key strategies to reduce the prevalence and the burden associated with mental illness in the population [41]. Over and above acknowledging the serious short- and long-term consequences of bullying in South African schools, there is a need to develop a comprehensive strategy to prevent and manage bullying in schools, which is preceded by the understanding of risk factors for being either a bullying perpetrator or victim. Such a strategy can inform the development of responsive interventions to curb this problem among learners. Based on previous empirical findings, we can expect that (H1) being a victim or perpetrator of bullying is associated with depression and anxiety and that (H2) there is an association between an unfavorable home environment and being bullied or bullying others. 

## 2. Methodology

### 2.1. Study Design

This study used a cross sectional quantitative survey design to screen for bullying behavior, which included perpetration of bullying, as well as being a victim of bullying.

This article, is part of a bigger mixed method study on the mental health impacts of bullying on perpetrators and victims among public high school learners in the Tshwane District of South Africa. This article focus on the profile of the learners that are perpetrators, victims, and perpetrator–victims, as well as the prevalence of the three categories of bullying in the sample.

### 2.2. Sampling

Multistage sampling was used to select schools and classes to participate in the study. Out of the fifteen public high schools in the specific Tshwane District township, seven were quintal 4, six were quintal 3, and two were quintal 1. Four quintal 4 schools were selected randomly from the list of all seven quintal 4 public high schools. Four quintal 3 schools were also randomly selected from the list of all six quintal 3 public high schools. The two quintal 1 public high schools were purposefully selected to ensure representation of all school quintals on the sample. Nine high schools participated in the study as one quintal 4 school declined to participate. One class per grade from grade 8 to 11 learner in the nine public high schools was selected randomly and all learners in the selected class were recruited to participate in the study.

### 2.3. Sample Size

The Raosoft sample size calculator was used for a population size of 20,000 (because the population size was unknown), with a 5% margin of error, a confidence level of 95%, and an estimated response rate of 50%, a minimum sample size of 377 was calculated, and the total sample size of 460 learners in grades 8 to 11 was reached. The age of the respondents ranged from 11 to 20 years with a mean age of 15 years.

### 2.4. Data Collection Tools

The Illinois Bully Scale was used to screen for bullying victimization and bullying perpetration. The Illinois Bully Scale is a validated tool [42,43,44,45]. It is an eighteen-item scale comprising of three subscales; namely victim subscale (4 questions with possible score ranging from 0–16), bully subscale (9 questions with possible score ranging from 0 to 38), and a fight subscale (5 questions). In this study, the victim and bully subscales were used. Adolescents were asked about the bullying actions in the last 30 days. The scale contains a 5-item Likert Scale with point values assigned as follows: Never = 0; 1 or 2 times = 1; 3 or 4 times = 2; 5 or 6 times = 3; and 7 or more times = 4. Subscale scores are computed by summing the respective items. A score of ≥1 on a victim scale is classified as a victim and a score of ≥1 on the bully scale is classified as a bully. Higher bully and victim scores indicate higher levels of bullying and victimization, respectively. Whilst international studies reported an internal consistency reliability Cronbach Alpha of 0.87 for the total subscale, 0.77 for the bully scale, and 0.71 for the victim subscale [43], a validation study conducted in a sub-Saharan African sample of secondary school learners found a Cronbach’s Alpha of 0.84 for the total scale, 0.79 for the bully subscale, and 0.78 for the victim subscale [45]. The Illinois Bully Scale asks about the bullying behaviors that occurred in the last 30 days.

The Patient Health Questionnaire (PHQ-9) was used to screen for depression symptoms. The PHQ-9 is a validated screening tool that has enjoyed global use because of its high specificity and high sensitivity levels [16,46,47] It is a nine-item scale with possible scores ranging from 0 to 27. The scale contains a 4-item Likert Scale with point values assigned as follows: Not at all = 0, several days = 1, more than half the days = 2, and nearly every day = 3. Subscale scores are computed by summing the respective items. A score of ≥8 was classified as a positive screen for depression as per the validation study conducted in South Africa [47]. A validity study on the PHQ-9 conducted in South Africa found a Cronbach Alpha internal reliability of 0.88 [47].

The Beck Anxiety Inventory (BAI) was used to screen for anxiety symptoms. The BAI is also a valid tool to screen for anxiety disorders with high sensitivity (82%) and specificity (80%) against the structured clinical interview for the DSM-5 at the score of ≥8) [48]. The BAI has a score range of 0–63. A score of ≥ 22 on the BAI is classified as positive screen for anxiety [48].

A researcher-developed questionnaire was used to collect socio-demographic data.

The socio-demographic data questions were informed by literature from a range of similar or related studies. Questions included age, gender, grade, school quintile, whether home environment was peaceful or there were fights, whether there was anyone that used alcohol frequently at home, whether they felt like their family loved and cared about them, whether they participated in sports or other mural activities at school, their academic performance, whether they had friends or not, and whether they stayed with their parents at home or not.

### 2.5. Recruitment

Permission to access the schools was sought and obtained from the Gauteng Department of Education. A list of all public high schools in the targeted township was obtained from the Tshwane South Department of Basic Education and selection of schools to participate from the list was conducted. School principals of the selected schools were contacted by email and by telephone to request permission for their schools to participate. School principals allocated Life Orientation Heads in their schools to work with the researcher. Meetings were held with the Life Orientation Heads for each school to brief them about the study and to seek their assistance. Class lists were obtained from the Life Orientation Heads and one class was selected randomly from the list of classes for each grade from grade 8 to 11. All learners in the selected class were recruited to participate and issued with consent forms together with parent information brochures explaining the study translated into SeTswana, SePedi, IsiZulu, and English, which are predominant languages in the area. A total of 759 learners were recruited to participate in the study and the response rate was 61%, which is above the 60% response rate which should be a goal of researchers [49].

### 2.6. Pilot Testing

The study was pilot tested among 5 high school learners aged between 14 and 19 years who were attached to a youth organization called LoveLife in the same township to determine if the questionnaire was easy to understand and the time it took to complete the questionnaire and any other issue that needed to be addressed before the actual study. Parental informed consent was obtained for the three learners that were under 18 years old, whilst the two that were 18 and 19 years signed consent after the procedure was explained. The feedback from the pilot test, which was on three socio-demographic questions, was integrated. In general, the learners found the questionnaire to be easy to complete and was completed between 12 and 15 min.

### 2.7. Data Collection Procedure

Data collection took place on dates and times agreed upon with a school. All learners whose parents had given consent were assembled in a class. The purpose of the study was explained and the potential participants were given an opportunity to ask questions. When they were ready, they signed the assent form, and then completed the demographic questionnaire, the PHQ-9 and the BAI. Data collection was paper- and pen-based. The questionnaire was in English, which is the medium of learning in the schools and clearly understood by the learners.

### 2.8. Data Analysis

Scores were added to determine the participants with symptoms of depression and anxiety and those without symptoms. A positive screen for depression symptoms was determined by a score of ≥8 on the PHQ-9 [47], whilst a score of ≥22 on the BAI was a positive screen for anxiety. A score of ≥1 on the Illinois Bully Scale bully subscale was classified as a bullying perpetrator, whilst a score of ≥1 on the victim subscale was classified as being a bullying victim [42].

The prevalence of bullying victimization was calculated by determining the percentage of participants that screened positive for bullying victimization on the Illinois Bully Scale victim subscale. The prevalence of bullying perpetration was calculated by determining the percentage of participants that screened positive for bullying perpetration on the Illinois Bully Scale bully subscale.

Questionnaire responses were captured on an Excel spreadsheet. Data were cleaned, coded, and imported into STATA version 14, StataCorp, College Station, TX, USA. Socio-demographic data were summarised using descriptive statistics. Bullying perpetration, victimization, and perpetrator-victim were depended variables. Depression symptoms, anxiety symptoms, and sociodemographic variables were independent variables.

Bullying perpetrator, bullying victim, and perpetrator–victim were the outcome variables. The association between each of the three outcome variables and each independent variable was determined using the Chi^2^ test. Association was determined by a *p*-value of <0.05. Logistical regression was used to determine the statistical significance of the observed associations between each of the independent variable and the dependent variables (*p*-value < 0.05).

### 2.9. Ethical Considerations

Ethical approval for the study was obtained from Sefako Makgatho Health Sciences University Research Ethics Committee (SMUREC/H/314/2020). Permission to conduct the study was granted by the Gauteng Department of Basic Education. The parent consent and brochures were translated to SePedi, IsiZulu, and SeTswana, which are languages predominantly spoken in the area. Only learners who agreed to participate, had written consent from their parents, and also signed the assent were allowed to participate. Prior arrangements were made with a social worker from the local LoveLife organization for referral of any learners showing signs of distress during or after participating in the study. Three learners were referred to the social worker through the school principal and involvement of parents, whilst five learners were brought to the attention of their school principals for assistance. All the referred learners either agreed to be referred or requested for assistance.

## 3. Results

Sixty-nine percent of the 460 respondents were females and were 31% males, with 25.4% in grade 8, 31.3% in grade 9, 19.4% in grade 10, and 23.9% in grade 11. The majority of respondents were attending quintal 4 schools (41.5%) followed by quintal 3 schools at 34.8% and lastly quintal 1 at 23.7%. There were no quintal 2 public high schools in the township. SePedi was the dominant home language (61.96%, *n* = 285), followed by IsiZulu (10.65%, *n* = 49), followed by SeTswana (8.48%, *n* = 39), followed by IsiNdebele (5.87%, *n* = 27), followed by South Sotho (5.43, *n* = 25), and the rest (7.61%) spoke Tsonga, Venda, Swati, and English. The majority of the respondents stayed with both parents (45.87, *n* = 211), 38.91% (*n* = 179) stayed with their mothers, 4.35%, (*n* = 20) stayed with their grandmothers, and 4.13% (*n* = 19) stayed with their fathers. The respondents’ birth positions were evenly distributed with 38.26% being first born, 29.57% being a middle child, and 25.87% last born. A small percentage (6.30%) were only child. With regards to family source of income, the majority of respondents reported that one of their parents was working (38.70%, *n* = 178), followed by the 27.17% (*n* = 125) who reported that both their parents were working, and followed by 7.83% (*n* = 36) who were receiving a child grant. A sizeable percentage of the sample screened positive for depression symptoms (45.00%, *n* = 207) and 21.30% (*n* = 98) screened positive for anxiety symptoms. Table 1 below shows the distribution of the socio-demographic characteristics of respondents.

### Involvement in Bullying Behavior

A total of 340 (73.91%) respondents were involved in bullying either as perpetrators (9.57%, *n* = 44), victims (21.96%, *n* = 101), or both (42.39%, *n* = 195). Bullying perpetration scores for the sample ranged between 0 and 22 with a mean score of 1.83 (SD = 3.15), whilst bullying victimization scores ranged from 0 to 20 with mean score of 2.72 (SD = 3.68).

Table 2 below shows that using the Pearson Chi^2^ test of association, being a perpetrator only was associated with anxiety symptoms (*p* < 0.037) and being from a home where alcohol was frequently used (*p* < 0.025). Being a victim only was associated with the perception that no one that loves and cares for the learner (*p* < 0.017), and being both a perpetrator and victim was associated with the school that they were attending (*p* < 0.021), home environment (*p* < 0.015), lack of family love and care (*p* < 0.012), as well as having depression (*p* < 0.000) and anxiety (*p* < 0.000) symptoms.

When the variables were controlled through logistic regression, depression and anxiety symptoms remained significantly associated with being a perpetrator–victim (*p* < 0.000, OR < 2.20, 95% confidence interval < 1.436487–3.371786 and *p* < 0.001, OR < 2.488, 95% confidence interval = 1.460963–4.238236, respectively). Perpetrator–victims were more likely to have someone using alcohol frequently at home (*p* < 0.012, coef 0.175796, 95% confidence interval 0.0393096–0.3157501). Absence of anxiety symptoms remained significantly associated with being a perpetrator only (*p* < 0.020, coef −1.708862, 95% confidence interval −3.159594–−0.2728312). School attended, home environment, and lack of family love and care were no longer significantly associated with being a perpetrator–victim, whilst having someone that uses alcohol frequently at home was no longer significantly associated with being a perpetrator, only when the confounders were controlled through logistic regression.

## 4. Discussion

This study may be the first to investigate both bullying perpetration and victimization in Tshwane District. The only study that examined bullying in this district was conducted almost two decades ago and focused on bullying victimization [12]. The 21.96% prevalence rate for bullying victimization found by this study is lower than what was found by other studies conducted in South Africa and other African countries [7,12,50]. However, these studies were limited to bullying victimization only. It is possible that some of the learners that were identified as victims could have been identified as perpetrators as well if they were asked about bullying perpetration and could therefore have been classified as perpetrator–victims. However, the 9.57% bullying perpetration prevalence that was found by this study is similar to other local and international studies [5,8,9,11]. The 42.39% perpetrator–victim prevalence is higher than what was reported by other studies in South Africa [9,11] and Nigeria [41]. The frequency of bullying incidents among high school learners that have been reported in various media platforms recently in this country [51] corroborate the findings by the current study. Similar to other studies that have found the prevalence rates of bullying perpetration to be lower than those reported for bullying victimization [5,8] the current study also obtained similar findings. The lower prevalence of bullying perpetration when compared to bullying victimization found by the current as well as other studies suggest that one perpetrator may bully more than one victim.The higher prevalence of perpetrator–victims found by this study suggests that more learners are assuming both the roles of being a victim and a perpetrator rather than being confined into one role, which on its own further contributes to the increasing prevalence of bullying in this target group. The strong association between being a perpetrator –victim and having both anxiety and depression symptoms found by this study is similar to findings by other local and international studies that have found an association between bullying and mental health problems [15,27]. Although some studies found an association between being a victim only and having depression and anxiety [31,32], this study did not find such an association. This is an unexpected finding as it goes against the findings by previous studies [4,17] and is also in contrast to relevant theories like the humiliation theory which posits that anger towards the perpetrator and pain because of humiliation can be internalized in the form of depression and anxiety [52]. This finding partially confirms the study hypothesis one (H1) in that depression and anxiety were associated with being a perpetrator–victim even though not with being a victim only. The findings of an association between having someone that drinks alcohol frequently at home and being a perpetrator–victim of bullying is similar to studies conducted among learners in high income countries [34]. These findings are not surprising as alcohol has frequently been linked to domestic violence [53] and domestic violence has been linked to bullying among children [54,55].

### 4.1. Strengths of the Study

The study measured three mental health constructs on the same sample, thus integrated links that run across depression, anxiety, and bullying behavior. The study also had a large sample size, with representation of all school quintiles and grades.

### 4.2. Limitations of the Study

The sample was drawn from one population group (Black), which compromises the ability to generalize the findings to other population groups.

This was a cross sectional study and therefore causal inference cannot be made.

The PHQ-9 and the BAI are screening tools that can only pick up symptoms of depression and anxiety but not confirm a diagnosis.

## 5. Conclusions

This study confirms the high rates of bullying among high school learners. The highest prevalence rate of perpetrator–victims suggests that learners are no longer confined to being either a victim or perpetrator but are assuming both roles, which may be depending on assumed power at a given time and situation. This further increases the proportion of learners that are exposed to being bullied. The findings that home situation and mental health symptoms are significantly associated with learners that are involved in bullying support the hypothesis of the study.

### Recommendations

The schools must have a specific strategy to identify bullying behavior and not only rely on isolated reported cases, as most victims tend not to report the behavior. Training of teachers about bullying and strategies that they can use to address bullying promise to be helpful as satisfaction after reporting bullying has been found to encourage learners to report bullying [56]. Screening and referral of learners for mental health problems should be institutionalized as part of the school health services. High risk approach anti-bullying interventions should prioritise learners from unstable home environments as well as those with mental health problems. There is a need to review and strengthen policies for bullying in schools to foster a balance between enforcing discipline and addressing the intrinsic drivers of bullying at individual level as well as the contextual drivers that underlie the problem of bullying.

## Figures and Tables

**Table 1 ijerph-20-04916-t001:** The socio-demographic characteristics of the respondents (*n* = 460).

Variable		Frequency (*n*)	Percentage (%)
Relationship with parents	Does not have parents	15	3.26
	Gets along well with parents	315	68.48
	Does not get along well with parents	17	3.70
	Gets along well with father but not with mother	17	3.70
	Gets along well with mother but not with father	96	20.87
	Total	460	100
Status of home environment	Family often gets into physical fights with each other	27	5.87
	Family often gets into verbal fights with each other	33	7.17
	Peaceful and close family	397	86.30
	Peaceful but not close family	3	0.65
	Total	460	100
Method used to discipline at home	No one disciplines me, I do as I please	40	8.70
	Privileges get taken away for some time	136	29.57
	Gets beaten up	42	9.13
	They shout at me	229	49.78
	They swear at me	12	2.61
	They don’t talk to me for some time	1	0.22
	Total	460	100
Frequent use of alcohol at home	No	328	71.30
	Yes, Mother	14	3.04
	Yes, Father	59	12.83
	Yes, Both parents	17	3.70
	Yes, other people I stay with	42	9.13
	Total	460	100
Lack of family love and care about them	No	31	6.74
	Yes	429	93.26
	Total	460	100
Presence of someone else that love and care about them	No	16	3.48
	Yes	444	96.52
	Total	460	100
Participation in sports or other activities at school	No	362	78.70
	Yes	98	21.30
	Total	460	100
Academic performance	Average	236	51.30
	High achiever	65	14.13
	Low achiever	134	29.13
	Really struggling academically	24	5.22
	Total	459	99.78
Number of friends at school	No friends	28	6.09
	Only 1 friend	95	20.65
	2 to 5 friends	274	59.57
	More than 5 friends	63	13.70
	Total	460	100
Method of travel to and from school	Designated school transport	26	565
	Public transport	134	29.13
	Dropped of and picked up by parent of relative	17	3.70
	Walk to and from school alone	112	24.35
	Walk to and from school with friends or siblings	171	37.17
	Total	460	100.00

**Table 2 ijerph-20-04916-t002:** Pearson Chi^2^ test of association between bullying behavior (perpetrator, victim, and perpetrator–victim) and socio-demographic variables.

Variable	Perpetrator (*p*-Value)	Victim (*p*-Value)	Perpetrator–Victim (*p*-Value)
Age	0.251	0.367	0.337
School that they were attending	0.573	0.380	0.021
School Quintal	0.658	0.577	0.851
Gender	0.860	0.860	0.987
Grade	0.112	0.599	0.095
Home language	0.234	0.587	0.498
Staying with at home	0.209	0.613	0.327
Birth position	0.998	0.845	0.600
Family income	0.838	0.234	0.786
Relationship with parents	0.659	0.645	0.248
Home environment	0.828	0.654	0.015
Discipline method at home	0.481	0.744	0.178
Anyone drinks alcohol frequently at home	0.025	0.175	0.032
Lack of family love and care	0.214	0.705	0.012
Presence of anyone else that loves and cares about them	0.478	0.017	0.809
Participation in sports or other activities at school	0.309	0.333	0.686
Academic performance	0.885	0.362	0.199
Number of friends at school	0.271	0.838	0.760
Method of travel to and from school	0.892	0.645	0.654
Depression symptoms	0.949	0.947	0.000
Anxiety symptoms	0.037	0.702	0.000

## Data Availability

Data may be available if requested according to data availability policies of Sefako Makgatho Health Sciences University.

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
