# Peer review of "The Profile of Bullying Perpetrators and Victims and Associated Factors among High School Learners in Tshwane District, South Africa"

_ijerph, 2023, doi:10.3390/ijerph20064916_

Round 1
Reviewer 1 Report
Congratulations to the authors for the interesting article which deepens the knowledge of bullying in adolescents.
There is an error in the method, as it starts describing the depression scale in the bullying section.
When describing the bullying scale, it would be convenient to indicate the time frame for which adolescents are asked when bullying has occurred, for example in the last two months, in the last year?
It would be necessary to add more detail of the scales used. For example between which number the scale moves, what does each one mean? This is important for example to determine how often the subjects are considered as perpetrators or victims.
The age data of the subjects should be indicated in the methodology section.
Some more complex analyses should be considered to try to explain the relationships between the variables. Such as regressions.
What was the reason for the large gender difference in the sample?
The discussion is very probing. Many relationships are made between variables that are not discussed. The discussion should be improved. References are also used in the discussion that have not been raised in the introduction.
Reviewer 2 Report
Review Report
Abstract:
· Four (4) instruments were used, and only three (3) were listed in the summary
· significance (p values) should not be reported in the logistic regression analysis result
Keywords:
· I would suggest simplified terminology, such as: Peer violence, perpetrators, bully victims, high school students, South Africa
Introduction
· Rearrange The introductory paragraph. Move the paragraph with data on the prevalence of violence and the paragraph on the roles of violence (victim, perpetrator) after the paragraph with definitions. The paragraph on the consequences of violence on mental health and the connection with other behaviors should be placed after the previously mentioned ones.
· Later, in the results section, the results of violence are indicated along with some variables from the questionnaire, which you named the sociodemographic questionnaire. The introduction lacks a review of previous knowledge about the variables involved (alcohol in the family, love and care in the family or from another important person) and violence.
· Hypotheses are missing at the end of the introductory section. Based on the presented scientific knowledge, set some expectations.
· The last sentence of the introductory paragraph: This article , which focuses on the profile of bullying perpetrators and victims , is part of a bigger mixed method study on the mentally health impacts of bullying on perpetrators and victims among public high school learners in the Tshwane District of South Africa , - It can be relocated in some of the next paragraphs, eg. Study desig. Also, put a fullstop at the end of that sentence.
Data collection tools
· The Illinois Bully Scale – the number of questions of each subscale and the possible range are indicated. Explain how the possible range was obtained and the range of responses offered. List here everything that the author of the scale mentioned. Later, roles in violence are mentioned - it is not clear whether this scale predicts roles. If yes, then state that in this paragraph. Please provide the Cronbach Alpha internal reliability of previous studies (if data are available for the South African population) and scale reliability of this study.
· Two questions adapted from the Cyberbullying and Online Aggression Survey Tool [34] were added to the Illinois Bully Scale - not clear how you added these two questions to The Illinois Bully Scale? If you modified the entire scale, describe the modification process in more detail. If you only added these two questions to the survey, since the results for these two questions are not shown, this should not be mentioned at all.
· The PHQ-9 that was used to screen for depression symptoms is a validated screening tool with high specificity and high sensitivity levels – this sentence is found in the paragraph on "The Illinois Bully Scale"(a). Move to the next paragraph (b).
· b) The Patient Health Questionnaire (PHQ-9) and c) The Beck Anxiety Inventory (BAI) - describe in more detail, such as the example described for the Illinois Bully Scales.
· d) The socio - demographic given questions similar or related studies. Describe in more detail. Which questions are included? What answers are offered?
Pilot testing
· The goal of pilot testing? (state the comprehensibility of the questions and the duration of completing the questionnaire).
· How many students participated in the survey?
· In relation to an informed parental consent, was it provided for their children in order to participate in the pilot study?
Data collection procedure
· Which was the form of the survey questionnaire? Did the students use pencil and paper to fill in the questionnaire?
· In which language was the questionnaire?
· Have you been informed on how many children were not given informed parental consent thus being excluded from the survey? Where were these children during the research? Were additional activities organized for them? If not, you are not obliged to indicate it. If yes, it is an example of good practice.
Data analysis
· First paragraph: Scores were added to determine the participants with symptoms of depression and anxiety and those without the symptoms . And positive screen for depression symptoms was determined by a score of ≥ 8 on the PHQ-9 ([39], while a score of ≥ 22 on the BAI was a positive screen for anxiety . A score of ≥ 1 on the Illinois Bully Scales the perpetrator subscale and on the victim subscale were classified as being a bully the perpetrator and a bullying victim respectively.
o state that the described procedure is for the purpose of creating dichotomous / categorical variables.
o according to whose criteria were the dichotomous variables created? If this criterion is part of the original scale, please indicate in the " Data collection tools" paragraph. In that case, state here how the dichotomous variables were created according to the original scale. If you carried out these procedures for the purposes of this scientific paper, please state the basis upon which these values were taken as critical for the formation of groups.
· Second paragraph: The prevalence of bullying victimisation was calculated by determining the percentage of participants that screened positive for bullying victimization on the Illinois Bullying Scale 's bullying victimization subscale . The prevalence of bullying perpetration was calculated by determining the percentage of participants that screened positive for bullying perpetration on the Illinois Bullying Scale 's bully subscale . The prevalence of perpetrator - victim was calculated by determining the percentage of participants that screened positive for both bullying victimization and bullying perpetration .
Was the described procedure conducted for the purpose of this paper? If yes, in parentheses, add a score that is the cut-off value for inclusion in the role of violence (eg > or < of). If the author of the scale stated so, insert this description in the description of the instrument and state here that you worked according to the author's recommendation.
· Describe how you converted the sociodemographic variables to dichotomous (those with multiple response options).
Results
· when you report the results of your study, the following should be stated: χ² = number , df = number , p</>0.05
· probability (p value) is usually stated as p<0.05 or p>0.05.
· When presenting the results of the logistic regression analysis, it is necessary to indicate whether the selected set of predictors significantly explains the criterion (Omnibus Tests of Model Coefficients ), how much the selected set of predictors explains the variances (Nagelkerke R²) as well as the number of participants correctly classified (Classification Table). In the description of predictors, probabilities should not be stated, but only those with significant contribution and with indicated direction (more or less). In parentheses there are the tests in SPSS, if that could help.
Discussion
· In the discussion paragraph, the association between roles in violence and the sociodemographic variables you used in the results is missing
Conclusion
· In the conclusion, briefly repeat the results with regard to the set hypotheses. Since there are no hypotheses at this time, it is not possible to check the completeness of the conclusion.
Recommendations
· The schools must have a specific strategy to identify bullying behaviours and note only rely on isolated reported cases - Specific strategies are mentioned. Can authors offer the examples? Explaining which strategies could be offered?
· In the introductory paragraph, the authors mention public policies, and here, the proposed guidelines aim at school responsibility. Connect the introduction and recommendations - if in one part the responsibility of public policy is mentioned, then in the second part also state the necessary recommendations to public policies. What is missing that could or should be stated in strategies and action plans? Thus, through this, the school can be mentioned as a place of comprehensive implementation of the program.
Generally
· Remove the space between comma and a number in square parentheses, related to references
· A few parentheses are redundant.

Author Response
We have responded to the extensive reviews, however, we have challenges in responding to the comments in yellow on the attached document. We believe that the paper is complete without the additional analysis required by the reviewer, which we are not able to comply with.

Reviewer 3 Report
This is an interesting article on the important issue. However, prior the publication there are some issues that should be sorted.
Introduction: add the prevalence in other countries, so that you can compare South African prevalence with it.
Please clearly describe the aim of the study.
Methods: Describe the response rate. also, there is a need for clear description of the outcome variable in the regression models.
Results: preform two multivariate logistic regression models with clearly defined outcome variables, present ORs and 95% Confidence Intervals. This way you can determine the profile of prepretrators and victims.
Discussion is insufficient. Please compare with other studies. Add limitations, such as that this was a cross-sectional study and that the causal relationship between the variables can not be established.
Round 2
Reviewer 1 Report
Congratulations to the authors for the improvements in the article.
Author Response
Thank you
Reviewer 2 Report
Review Report No.2
I am grateful to the authors for considering my review. Great job. I agree with the author's opinion on my comments which they marked in yellow. Working in different statistical programs does not give the same presentation of the conducted analyses.
Some of my comments on the latest version of the paper are as follows:
1. After line 49: apply the rule about referencing in the text: References must be numbered in order of appearance in the text and listed individually at the end of the manuscript.
2. e.g. lines 47 or 59: Reference numbers should be placed in square brackets [ ] and placed before the punctuation; for example [1], [1–3] or [1,3].
3. Line 81: delete the extra bracket
4. Line 253: 460 LEAR69%- is this a mistake?
5. Line 258: remove the space after the parenthesis
6. Lines 279 to 284: probability (p value) is usually stated as p<0.05 or p>0.05
7. The text lacks a note which results refer to which table.
8. You can write hypotheses at the end of the Introduction paragraph. Proposal: Based on previous empirical findings, we can expect that (H1) being a victim or perpetrator of bullying is associated with depression and anxiety and that (H2) there is an association between unfavorable home environment and being bullied or bullying others.
9. Line 292: the second parenthesis is missing
10. Full stops are missing at the end of the sentence (e.g. lines 63, 92, 182, 197, 232, 320, 341, 343).
11. Excess spaces (e.g. line 309)
12. Line 302: delete the comma after reference 52
13. Line 305: delete the full stop in the middle of the sentence
14. Based on the statement in line 323-324 (Though some studies found an association between being a victim only and having depression and anxiety [33,34], this study did not find such an association.), in the Conclusion lacks the finding that hypothesis 1 is partially confirmed.
15. Align decimal numbers throughout the text (full stop or comma)
I wish You all the best in Your further scientific work!
Author Response
The responses to reviewer 2 are attached
